# The Impact of Ag Nanoparticles and CdTe Quantum Dots on Expression and Function of Receptors Involved in Amyloid-β Uptake by BV-2 Microglial Cells

**DOI:** 10.3390/ma13143227

**Published:** 2020-07-20

**Authors:** Katarzyna Sikorska, Iwona Grądzka, Iwona Wasyk, Kamil Brzóska, Tomasz M. Stępkowski, Malwina Czerwińska, Marcin K. Kruszewski

**Affiliations:** 1Centre for Radiobiology and Biological Dosimetry, Institute of Nuclear Chemistry and Technology, Dorodna 16, 03-195 Warsaw, Poland; i.gradzka@ichtj.waw.pl (I.G.); iwonawasyk@gmail.com (I.W.); k.brzoska@ichtj.waw.pl (K.B.); t.stepkowski@cent.uw.edu.pl (T.M.S.); m.wasilewska@ichtj.waw.pl (M.C.); m.kruszewski@ichtj.waw.pl (M.K.K.); 2Department of Molecular Biology and Translational Research, Institute of Rural Health, Jaczewskiego 2, 20-090 Lublin, Poland

**Keywords:** BV-2 microglial cells, amyloid-β uptake, scavenger receptors, metallic nanoparticles

## Abstract

Microglial cells clear the brain of pathogens and harmful debris, including amyloid-β (Aβ) deposits that are formed during Alzheimer’s disease (AD). We studied the expression of Msr1, Ager and Cd36 receptors involved in Aβ uptake and expression of Cd33 protein, which is considered a risk factor in AD. The effect of silver nanoparticles (AgNP) and cadmium telluride quantum dots (CdTeQD) on the expression of the above receptors and Aβ uptake by microglial cells was investigated. Absorption of Aβ and NP was confirmed by confocal microscopy. AgNP, but not CdTeQD, caused a decrease in Aβ accumulation. By using a specific inhibitor—polyinosinic acid—we demonstrated that Aβ and AgNP compete for scavenger receptors. Real-time PCR showed up-regulation of *Cd33* and *Cd36* gene expression after treatment with CdTeQD for 24 h. Analysis of the abundance of the receptors on the cell surface revealed that AgNP treatment significantly reduced the presence of Msr1, Cd33, Ager and Cd36 receptors (6 and 24 h), whereas CdTeQD increased the levels of Msr1 and Cd36 (24 h). To summarize, we showed that AgNP uptake competes with Aβ uptake by microglial cells and consequently can impair the removal of the aggregates. In turn, CdTeQD treatment led to the accumulation of proinflammatory Cd36 protein on the cell surface.

## 1. Introduction

Alzheimer’s disease (AD) is one of the most prevalent neurodegenerative disorders in the world. The main feature of the disease is the presence in the brain of extracellular amyloid-β (Aβ) deposits, called plaques, and intracellular neurofibrillary tangles, which are built by Tau protein. Plaques are formed as a result of the defective cleavage of amyloid precursor protein [1,2]. The Aβ deposits are toxic to neurons as they generate oxidative stress and induce proinflammatory responses in the brain [2]. It is generally agreed that there is currently no effective cure for AD [3]. Drugs, which are available for patients, target the symptoms of AD and do not have any impact on the inhibition or diversion of the disease progress. Therefore, there is a need for new drugs which are able to prevent plaque formation.

One of the promising new trends in drug development is the use of nanoparticles (NP) for diagnosis and/or treatment of the disease [4,5,6,7]. For example, curcumin-conjugated superparamagnetic iron oxide NP [8] and AuNP–antibody complexes [9] were used to visualize Aβ plaques in AD patients. NP may also affect Aβ aggregation due to physical interaction with the peptide. It was shown that PEGylated poly((hexadecyl cyanoacrylate)-co-methoxypoly(ethylene glycol) cyanoacrylate) P(HDCA-co-MePEGCA) NP were able to keep Aβ monomers and soluble oligomers in solution [10]. In this way, they can inhibit the formation of new Aβ plaques. Moreover, N-acetyl-L-cysteine-capped cadmium telluride quantum dots (CdTeQD) were able to disaggregate existing Aβ plaques or fibrils [11].

Apart from these complex nanostructures which are intentionally engineered for medical use, a number of simple NP commonly used in the industry, such as TiO_2_NP or SiO_2_NP, are present in the human environment. One of the most abundant due to their antibacterial, antifungal and antiviral properties are silver NP (AgNP), widely used in medical products, cosmetics, textiles, etc. [12,13]. CdTeQD, due to their unique photo-physical properties, are widely used in electroluminescence devices but also in biomedicine as markers in molecular bioimaging techniques [5]. All of these NP may penetrate the brain and cause adverse effects on human health.

In the brain, microglia constitute a natural defense against pathogens and other toxic factors, including Aβ. Microglia are able to translocate to the site of their accumulation and remove them through phagocytosis [14,15,16,17]. Interaction of Aβ with microglia takes place via specific receptors on the cell surface, such as formyl peptide receptor 1 (Fpr1), Cd36 molecule (also known as a scavenger receptor class B member 3), advanced glycosylation end-product specific receptor (Ager), Cd47 molecule, Cd14 molecule, toll-like receptors (Tlrs), α6β integrin and macrophage scavenger receptor 1 (Msr1) [14,15,18,19,20,21,22]. In our previous study, we showed that CdTeQD and AgNP were toxic to microglial cells; moreover, AgNP efficiently blocked Aβ uptake by microglia, thus impairing their function [23]. Hence, it is likely that AgNP block or somehow interact with the activity of these receptors. Thus, in this work, we focused on the effects of AgNP and CdTeQD on the activity of three main receptors involved in Aβ uptake: Msr1 (also known as scavenger receptor class A member 1, Scara-1), Ager (also known as a receptor for advanced glycation end products, Rage) and Cd36 and Cd33 molecule protein (also known as a sialic acid binding Ig-like lectin 3). The function of Msr1 in the presence of NP was investigated in more detail.

## 2. Materials and Methods

### 2.1. Amyloid β

Human Aβ (1–42), unlabeled and labeled with HiLyte™ Fluor 488 (ex/em 503/528 nm), was purchased from AnaSpec Inc. (Seraing, Belgium). This was a mixture of monomers and amorphous aggregates of various size and shape [23]. The Aβ was dissolved in deionized water (500 µM) and frozen in aliquots at −80 °C for further use.

### 2.2. Cell Culture

BV-2 mouse microglial cells were purchased from the Biological Bank and Cell Factory IRCCS San Martino (Genoa, Italy). The cells were maintained in RPMI-1640 medium supplemented with 10% fetal bovine serum (FBS) (Biological Industries, Kibbutz Beit Haemek, Israel) and 2 mM L-glutamine. The cells were kept at 37 °C in a humidified 5% CO_2_ incubator and passaged every two days.

### 2.3. Nanoparticle Preparation and Characterization

AgNP of nominal size 20 nm and CdTeQD of nominal size 3.8 nm were purchased from Plasmachem GmbH (Berlin, Germany) and prepared as previously described [23]. In brief, 2 mg of NP was suspended in 800 μL of deionized water and sonicated (10 min) on ice at a radiant energy density of 4.2 kJ cm^−3^. Immediately after sonication, 100 µL of 15% BSA (bovine serum albumin) followed by 100 µL of 10× PBS (phosphate-buffered saline) were added to obtain an NP stock suspension (2 mg mL^−1^) in PBS with 1.5% BSA, pH 7.1. Full characterization of the NP used in this study is provided in [23].

### 2.4. Visualization of Aβ and NP Uptake and Localization

Aβ and CdTeQD uptake was visualized with confocal microscope Nikon A1 (Nikon, Tokyo, Japan) equipped with NIS-elements AR 4.13.04 software (Nikon, Tokyo, Japan). Microglial cells were seeded on chamber slides (5 × 10^4^ per slide) and left to grow overnight. The next day, the cells were incubated with Aβ-HiLyte Fluor 488 (0.1 μM) for 0, 5, 15, 30 or 60 min, or with CdTeQD (10 μg mL^−1^) for 2 h, then the fluorescence of selected cells was measured. In order to localize the absorbed Aβ or CdTeQD within cells, the NP were co-incubated with a marker of lysosomes (LysoTracker^®^ Red, ex/em 577/590 nm, Thermo Fisher Scientific, Waltham, MA, USA) or a marker of mitochondria (MitoTracker^®^ Red, ex/em 579/599, Thermo Fisher Scientific, Waltham, MA, USA). AgNP uptake was visualized after 2 h incubation using an inverted light microscope (Nikon Eclipse Ti, Tokyo, Japan).

### 2.5. Measurement of Aβ and NP Uptake by Flow Cytometry

Flow cytometry measurements were performed with the use of the BD-LSR Fortessa (BD Biosciences, San Jose, CA, USA) flow cytometer equipped with BD FACSDiva 8.0.1 software (BD Biosciences, San Jose, CA, USA). Microglial cells were grown overnight in 6-well plates to a density of ca. 1.35 × 10^5^ cells/well. For examination of the Aβ and NP uptake, the cells were treated with Aβ (0.1 μM) and/or CdTeQD (0.1 or 10 μg mL^−1^) or AgNP (5 or 50 μg mL^−1^) for 2 h. Then, they were harvested by trypsinization, washed with the growth medium containing 10% FBS and resuspended in the medium containing 2% FBS. Aβ and CdTeQD uptake was estimated on the basis of their mean fluorescence intensity (at 528 and 655 nm, respectively), while AgNP accumulation in the cells was measured as an increase in the mean side scatter (SSC) value [24].

### 2.6. Assessment of the Role of Scavenger Receptors Class A in Aβ and NP Uptake

To assess the contribution of scavenger receptors class A in the Aβ and NP uptake process, the cells were preincubated for 0.5 h with polyinosinic acid (PA, 50 μg mL^−1^), a specific inhibitor of this type of receptor, and then the Aβ and NP uptake was measured by flow cytometry, as described above.

The impact of AgNP and CdTeQD on the expression of cell surface receptors was determined also by flow cytometry, by the use of the following specific fluorescein isothiocyanate (FITC)-conjugated antibodies: anti-Cd36 and anti-Msr1 (Life Technologies, Carlsbad, CA, USA), anti-Ager (LSBio LifeSpan BioSciences, Seattle, WA, USA) and anti-Cd33 (Bioss Antibodies, Woburn, MA, USA). Microglial cells were treated with AgNP (5 or 50 μg mL^−1^) or CdTeQD (0.1 or 10 μg mL^−1^) for 6 or 24 h. One hour before the cells were harvested, a DNA dye Hoechst 33,258 (Sigma-Aldrich, Saint Louis, MO, USA, 10 mg mL^−1^) was added to enable identification of the cells. At the end of incubation, the cells were detached from the plate, spun down (200× *g*, 4 °C, 5 min), resuspended in 100 μL of the medium and a specific FITC-conjugated antibody was added for 30 min at room temperature. After incubation with antibody, the cell suspension was diluted with the culture medium without FBS and the fluorescence intensity of the cells was measured by flow cytometry.

### 2.7. Real-Time PCR

The RT PCR method was used to analyze the expression of genes coding receptors involved in Aβ clearance by microglia, i.e., *Cd33, Cd36, Ager* and *Msr1*. The cells were treated with AgNP (5 or 50 μg mL^−1^) or with CdTeQD (0.1 or 10 μg mL^−1^) for 6 and 24 h. RNA was isolated from the samples using ReliaPrep^TM^RNA Cell Miniprep System (Promega, Madison, WI, USA), according to the manufacturer’s protocol. To assess concentration and purity of RNA, every sample was diluted in TE buffer (pH 8.0) and the absorbance at 260 nm was measured using a Cary 50 UV-Vis spectrophotometer (Varian, Palo Alto, CA, USA). For real-time PCR analysis, 1 μg of total RNA was converted to cDNA in a 20 μL reaction volume, using the High Capacity cDNA Reverse Transcription Kit (Thermo Fisher Scientific, Waltham, MA, USA), following the manufacturer’s instructions. The cDNA was diluted with 130 μL nuclease-free H_2_O and used for the real-time PCR reaction. The assay was performed in a 20 μL reaction mixture consisting of 5 μL cDNA, 4 μL nuclease-free H_2_O, 10 μL TaqMan Universal Master Mix II no UNG (Thermo Fisher Scientific, Waltham, MA, USA) and 1 μL of TaqMan Gene Expression Assay (Thermo Fisher Scientific, Waltham, MA, USA). The following assays were used: Mm00446214_m1 (*Msr1*); Mm01134790_g1 (*Ager*); Mm01135198_m1 (*Cd36*); Mm00491152_m1 (*Cd33*); Mm99999915_g1_ (*Gapdh)*. Available information on each assay can be found at the Thermo Fisher Scientific website. The mRNA expression was calculated by the ΔΔ Ct method relative to a reference gene, *Gapdh.* The reaction was conducted using Applied Biosystem 7500 Real-Time PCR System (Foster City, CA, USA). Reaction conditions were as follows: 95 °C for 10 min, and then 40 cycles of 95 °C for 15 s, followed by 60 °C for 1 min. Data acquirement and analysis were performed using the Relative Quantification Software version 3.2.1-PRC-build1, (Thermo Fisher Scientific, Waltham, MA, USA.)

### 2.8. Statistical Analysis

Data are shown as means ± SD of at least three independent experiments. Statistical differences between means were determined by one-way ANOVA, followed by Tukey’s multiple comparison test or by paired *t*-test. Statistical analyses were performed using Graphpad 5.0 (San Diego, CA, USA). Differences were considered statistically significant when the *p* value was <0.05.

## 3. Results

### 3.1. The Uptake of Aβ and NP by Microglial Cells

Uptake of Aβ by microglia was visible as soon as 5 min after the addition of Aβ, and its accumulation constantly increased during the 2-h observation period (Figure 1). Similarly, the cells were able to uptake CdTeQD and AgNP (Figure 2, Figure 3, respectively). Aβ and CdTeQD preferentially accumulated in the cytoplasm (Figure 1 and Figure 2).

Localization of Aβ in the cells’ organelles was visualized using fluorescent markers specific for mitochondria or lysosomes and confirmed by calculation of the Pearson’s correlation coefficient (PCC) of both fluorescence distributions [25]. Colocalization between Aβ and mitochondria (Figure 4) was poor (PCC = 0.14 ± 0.16, for 10 randomly selected cells). On the contrary, localization of Aβ and the lysosome marker revealed significant similarities (Figure 5) (PCC = 0.694 ± 0.06, for 14 randomly selected cells).

### 3.2. Analysis of Gene Expression in BV-2 Cells Treated with Aβ or NP

RT PCR was used to assess the impact of NP on the expression of genes coding receptors of interest, i.e., *Msr1*, *Cd36*, *Ager* and *Cd33*. After 6 h of treatment, no changes were seen (Appendix A). Similarly, no changes were observed after 24 h treatment in cells incubated with AgNP (5 or 50 μg mL^−1^). In case of treatment with CdTeQD at a low concentration (0.1 μg mL^−1^), a slight, but statistically significant, decrease in *Cd33* expression took place, whereas at the concentration of 10 μg mL^−1^, a slight but significant increase in this gene’s expression was observed. Interestingly, treatment with 10 μg mL^−1^ CdTeQD resulted in an almost four-fold increase in expression of *Cd36* gene (Table 1).

Data were compared to untreated controls. Results are means of three independent experiments; RqMin and RqMax are also shown. Statistically significant effects (*p* < 0.05) are given in boldface. Rq (relative quantification) is a relative change in a gene expression; see Materials and Methods section.

### 3.3. Role of Scavenger Receptors Class A in Aβ and NP Uptake

To assess the potential role of scavenger receptors class A in Aβ, AgNP and CdTeQD uptake, the cells were preincubated with scavenger receptors class A inhibitor, polyinosinic acid, and then incubated with Aβ or NP. Further analysis of fluorescence (Aβ and CdTeQD) or size scatter (AgNP) revealed a marked shift in the signal intensity from Aβ and AgNP. On the contrary, no statistically significant shift in the signal intensity was observed for CdTeQD (Figure 6, Table 2).

The abundance of Msr1, Cd36, Ager and Cd33 proteins on the surfaces of the BV-2 cells was measured cytometrically using FITC-labeled specific antibodies after 6 and 24 h of incubation with NP. Interestingly, 6 h treatment with 50 µg mL^−1^ AgNP resulted in a decrease in the abundance of all tested receptors on the cell surface (Figure 7). In the case of Msr1, this effect was also observed after 24 h incubation and for 6 h treatment with 5 µg mL^−1^ AgNP. On the contrary, treatment with CdTeQD (10 µg mL^−1^, 24 h) resulted in an increase in the abundance of Msr1 and Cd36 receptors on the BV-2 cell surface.

## 4. Discussion

In our previous work, we have shown that AgNP at 50 μg mL^−1^ significantly diminished the accumulation of Aβ in the BV-2 cells, whereas CdTeQD (10 μg mL^−1^) did not have any impact on Aβ uptake [23]. In this work, we investigated the effect of AgNP and CdTeQD treatment on the expression and function of selected proteins involved in Aβ clearance by microglial cells, namely the following cell surface receptors: Msr1, Ager, Cd36, Cd33. Msr1 belongs to the subfamily of scavenger receptors class A and participates directly in the process of Aβ phagocytosis [22,26,27]. Msr1 deficiency impairs the clearance of soluble Aβ by mononuclear phagocytes and accelerates Alzheimer’s-like disease progression [26]. Ager is expressed on endothelial cells and microglia. Aβ binding to Ager induces microglial activation and chemotaxis, which leads to microglial accumulation around Aβ aggregates [28,29]. Cd36 molecule belongs to the family of scavenger receptors class B and is responsible for long fatty acid uptake, but it also participates in Aβ uptake. Cd36 receptor mediates Aβ-induced secretion of cytokines, chemokines and reactive oxygen species by microglia. Thus, it plays a key role in inflammatory events associated with AD [30,31]. Cd33 protein was shown to inhibit Aβ 1-42 clearance by microglial cells. The mechanism of this phenomenon is associated with sialic acid binding but still remains obscure. It was proven that the *Cd33* gene expression is increased in AD [32,33].

In our work, the internalization and intracellular localization of Aβ by BV-2 cells was confirmed using confocal microscopy and flow cytometry. Internalization of Aβ was very fast and positively correlated with time of incubation (Figure 1B). The similar kinetics of the Aβ uptake by human and mouse microglia was reported by others [31,34]. Aβ was accumulated in the cytoplasm, not in the nuclei (Figure 1A), and colocalized with lysosomes (Figure 5), which is also consistent with earlier reports [18,35,36,37,38]. The uptake and distribution of CdTeQD was also visualized by confocal microscopy (Figure 2) and was similar to that demonstrated by others [39]. The uptake of AgNP was proven using light microscopy (Figure 3) and it was in accordance with the uptake demonstrated in mouse microglial N9 cells [40].

In the previous publication, we showed that the presence of AgNP, but not CdTeQD, inhibited Aβ uptake by BV-2 cells [23]. Here, we attempt to examine the cellular mechanisms responsible for this observation. Analysis of RNA isolated from AgNP-treated cells revealed no effect of AgNP on the expression of the genes of interest (Table 1). However, analysis of the abundance of these receptor proteins on the cell surface revealed that 6 h treatment with a high concentration of AgNP (50 µg mL^−1^) resulted in decreased numbers of all tested receptor proteins, as detected by specific antibody (Figure 7). Since all receptor proteins were affected, the mechanism must be general for cell membrane proteins. Although we did not test it, the diminished availability of lipid rafts as a result of AgNP endocytosis comes to mind as a mechanism responsible for the decreased abundance of membrane receptors in AgNP-treated cells. The lack of effects after 24 h incubation is likely due to the receptor recycling after internalization.

Although the AgNP-induced decrease in the abundance of membrane receptors was common for all receptors tested, it was the most pronounced for Msr1 receptor. Similar to other tested receptors, 6 h treatment with a high concentration of AgNP resulted in a decrease in abundance of this receptor on the cell surface. However, contrary to other receptors, this effect was pronounced for 24 h. Moreover, treatment with a low AgNP concentration also resulted in a decrease in the abundance of Msr1 receptor after 6 h, which suggests that an additional mechanism is involved. The existence of two independent mechanisms of AgNP uptake was earlier proposed for the uptake of AgNP by human J774A.1 macrophages [41]. Thus, we decided to use Msr1 receptor inhibitor to determine whether there is any direct interaction between AgNP and the receptor. Indeed, pretreatment with Msr1 receptor inhibitor almost completely inhibited the uptake of both Aβ and AgNP (Figure 6). Our results indicate that Aβ and AgNP enter the cells through the same receptor pathway. Consistent with this, other authors reported that NP, including AgNP, are transported into the cell through scavenger receptor dependent phagocytosis [40,42,43,44,45]. Additional proof for the involvement of scavenger receptors in AgNP uptake was a clear decrease in the amount Msr1 receptors, estimated by specific antibodies on the cell surface in AgNP-treated cells (Figure 7). Though the exact mechanism of AgNP binding to Msr1 receptor is not known, we speculate that it might be unspecific binding to thiol (-SH) groups. AgNP have a strong affinity for cysteine tiol (-SH) groups of proteins [46] that are abundant in Msr1 scavenger receptor [47].

Thus, the observed inhibition of Aβ uptake by AgNP [23] is likely due to the competition for the same surface receptors. In our previous work, we also demonstrated that pretreatment with CdTeQD has no effect on Aβ uptake [23]. This observation was also confirmed here, as, despite the apparent uptake of CdTeQD by BV-2 cells, pretreatment with Msr1 receptor inhibitor did not affect CdTeQD uptake (Figure 6). Treatment with CdTeQD also had negligible effects on the expression of *Cd33, Ager* and *Msr1* genes (Table 1). However, treatment with CdTeQD resulted in an almost four-fold increase in *Cd36* expression on the mRNA level (Table 1) and an over 1.5-fold increase in the Cd36 protein on the cell surface (Figure 7D). As already mentioned, Cd36 receptor is involved in Aβ uptake, but it also mediates the Aβ-induced microglia inflammatory response. Thus, stimulation of its expression might have detrimental effects when CdTeQD interact with microglia.

Collectively, we have documented the negative effects of AgNP and CdTeQD on microglial cells. AgNP compete with Aβ for scavenger receptors and impair its clearance by BV-2 microglia. Although CdTeQD increase the expression of Cd36, both on a mRNA and protein level, this might enhance the microglia inflammatory response mediated by the presence of Aβ. Thus, both NP may have adverse effects in Alzheimer’s disease. It seems necessary to coat nanoparticles in order to reduce their toxicity, and they should be examined in detail before being used for the diagnosis or treatment of this neurodegenerative disorder.

## Figures and Tables

**Figure 1 materials-13-03227-f001:**
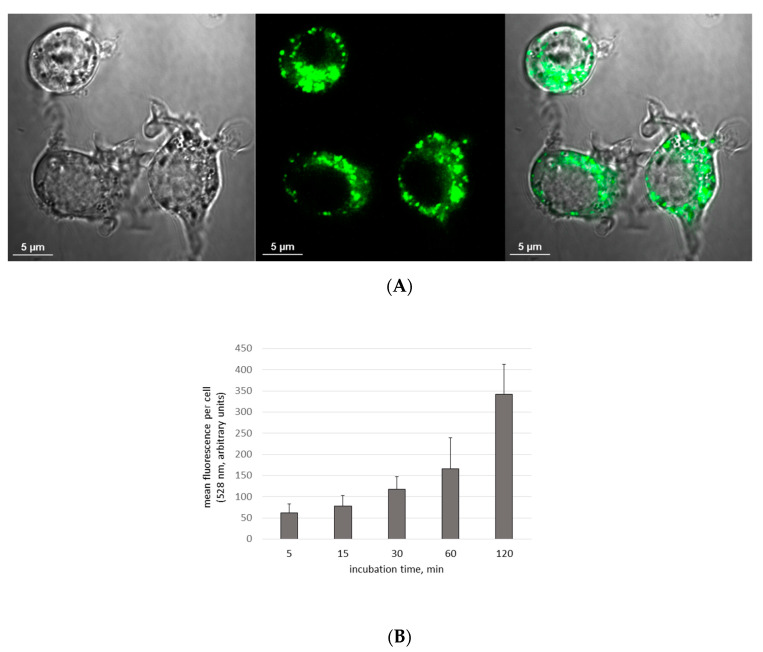
Uptake of Aβ by BV-2 cells. (**A**) Confocal microscopy image after 60 min incubation with Aβ-HiLyte Fluor 488. Magnification 100×. From left to right: DIC (differential interference contrast) microscopy, Aβ-HiLyte Fluor 488 fluorescence, merged. (**B**) Kinetics of Aβ uptake by microglial cells. BV-2 cells were incubated with Aβ-HiLyte Fluor 488 (0.1 μM) for 0, 5, 15, 30, 60 and 120 min. Fluorescence intensity at 528 nm was measured in six randomly selected cells using confocal microscope. Mean ± SD.

**Figure 2 materials-13-03227-f002:**
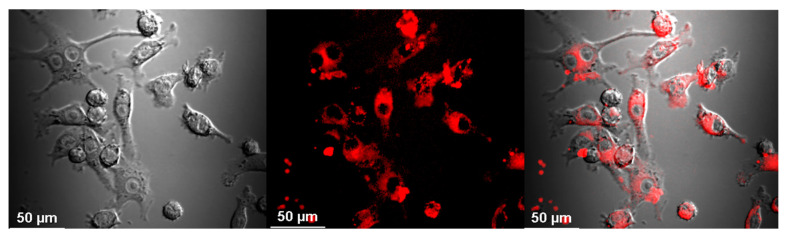
CdTeQD uptake by BV-2 cells. Confocal image of the cells incubated with CdTeQD for 2 h 45 min. Magnification 40×, from left to right: DIC microscopy, CdTeQD fluorescence, merged.

**Figure 3 materials-13-03227-f003:**
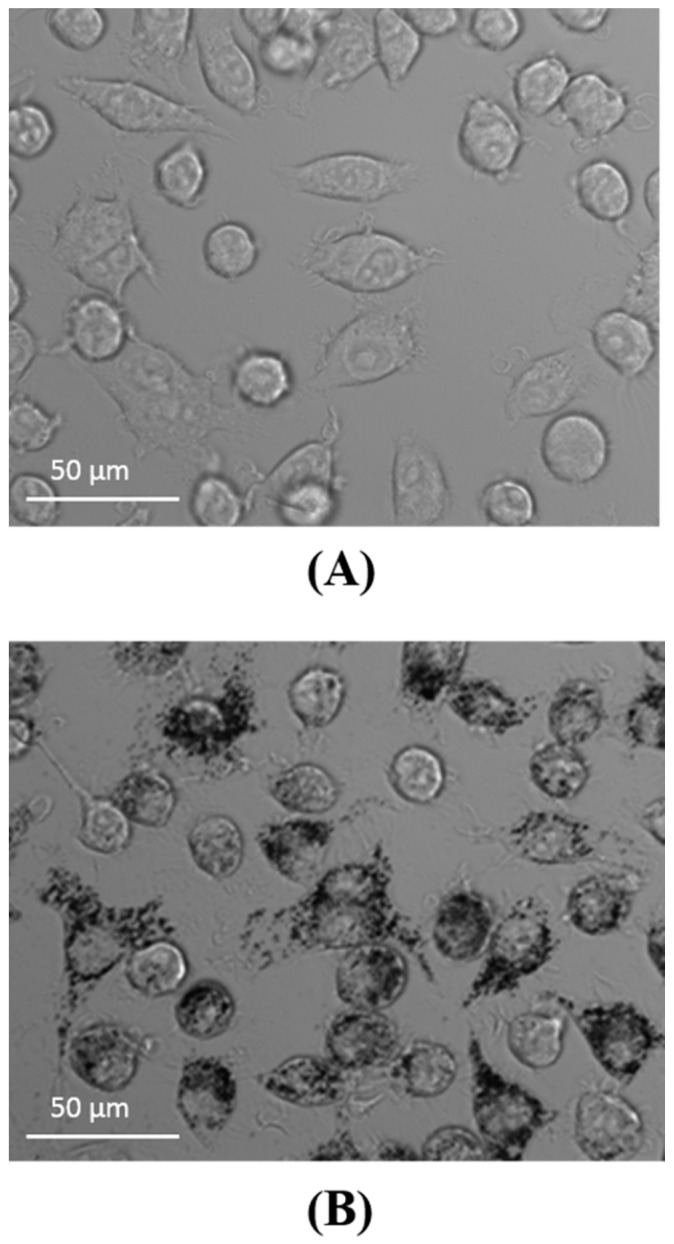
AgNP uptake by BV-2 cells. (**A**) Control cells. (**B**) The cells incubated with AgNP for 2 h, light microscopy image, magnification 40×. AgNP are visible as black points.

**Figure 4 materials-13-03227-f004:**
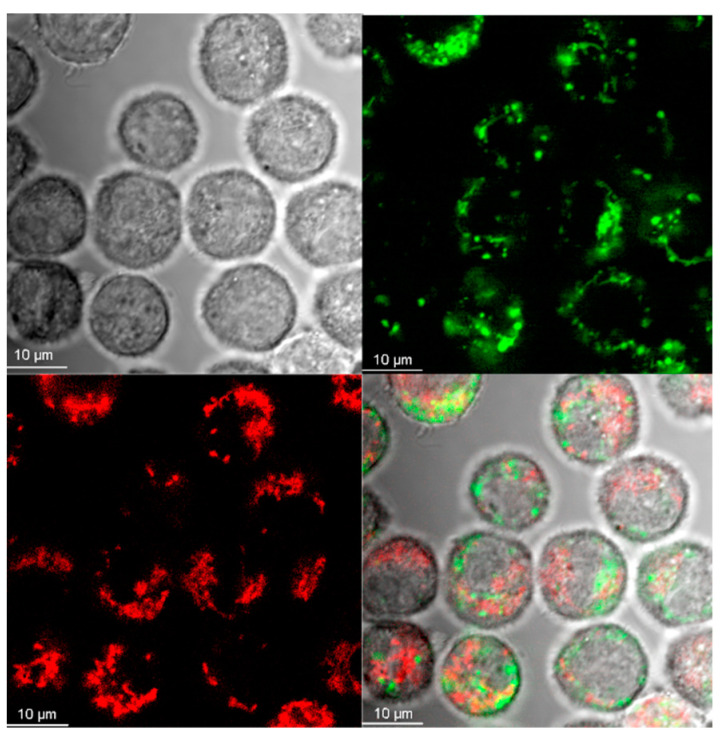
Colocalization of Aβ-HiLyte Fluor 488 (green) and mitochondrial marker MitoTracker Red (red) in BV-2 cells. Confocal microscope, magnification 100×.

**Figure 5 materials-13-03227-f005:**
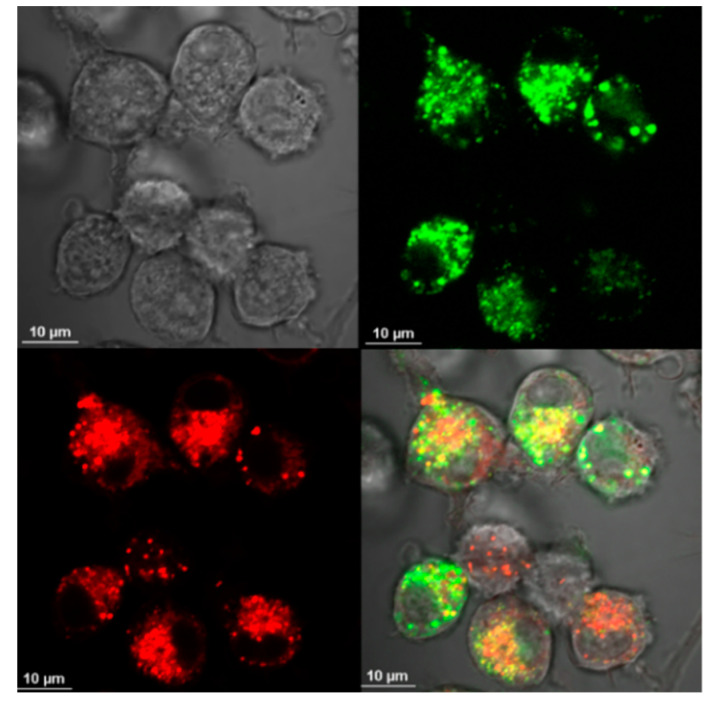
Localization of Aβ-HiLyte Fluor 488 (green) and lysosome marker LysoTracker Red DND-99 (red) in BV-2 cells. Confocal microscope, magnification 100×.

**Figure 6 materials-13-03227-f006:**
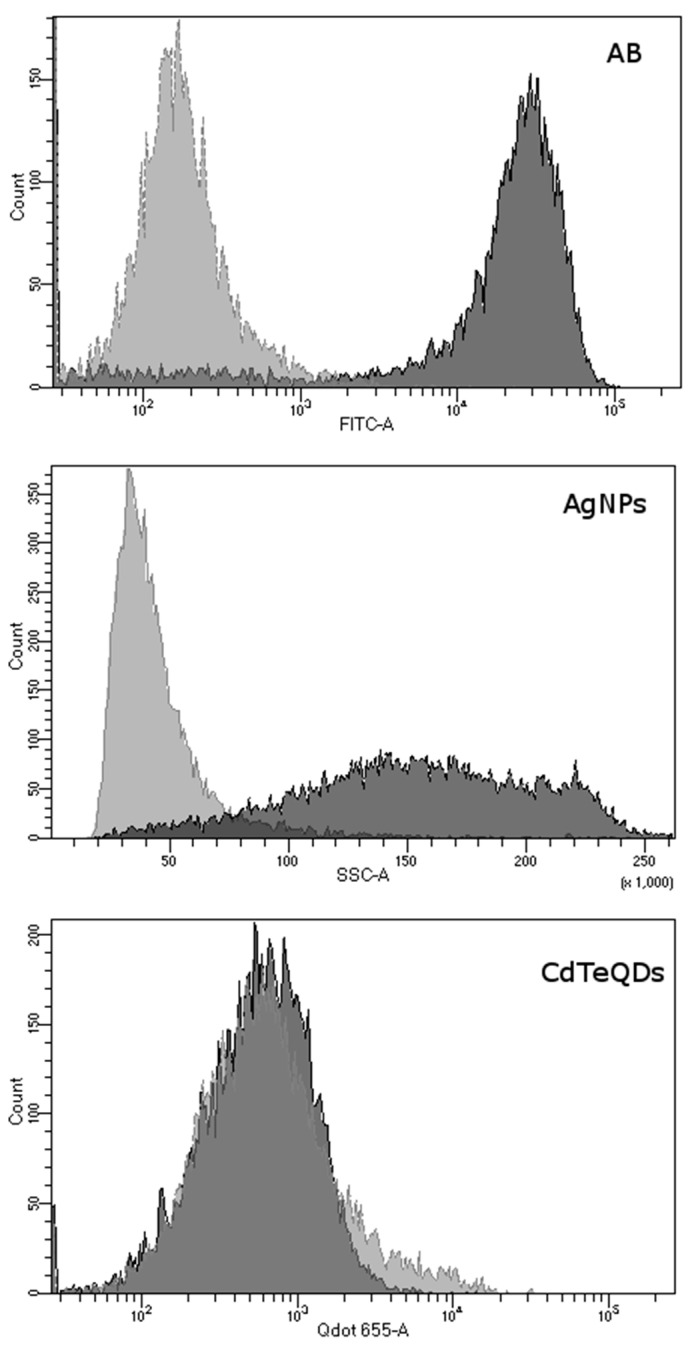
Uptake of Aβ, AgNP and CdTeQD by BV-2 cells preincubated with polyinosinic acid (PA). The cells were pretreated (or not) with PA, 50 μg mL^−1^ for 30 min, then Aβ-HiLyte Fluor 488 (0.1 μM), AgNP (50 μg mL^−1^) or CdTeQD (10 μg mL^−1^) were added for 2 h. Light grey colored cells were pretreated with PA; dark grey colored cells were not treated with PA. Exemplary results of one of the experiments. Numerical data provided in Table 2.

**Figure 7 materials-13-03227-f007:**
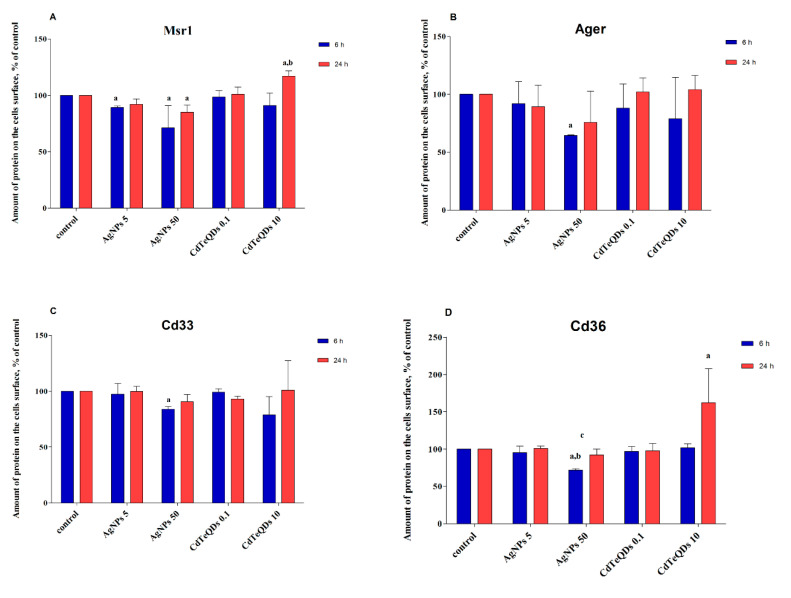
The abundance of Msr1 (**A**), Ager (**B**), Cd33 (**C**)or Cd36 (**D**) receptors on the BV-2 cells surface by flow cytometry. The cells were treated with CdTeQD (0.1 or 10 µg mL^−1^) or AgNP (5 or 50 µg mL^−1^) for 6 or 24 h. Mean ± SD, *n* = 3. ‘a’ denotes statistically significant difference from unexposed control, ‘b’ denotes statistically significant difference between low and high concentration of NP, ‘c’ denotes statistically significant difference between 6 and 24 h treatment, two-way ANOVA, post-hoc Tukey’s test, *p* < 0.05.

**Table 1 materials-13-03227-t001:** The *Msr1*, *Cd36*, *Ager* and *Cd33* gene expression of BV-2 cells after treatment with NP for 24 h (real-time PCR).

Sample	Gene	Mean Rq (Relative Quantification) Value	Rq Min	Rq Max	*p* Value
**Control**	*Ager*	1	0.913	1.096	1
	*Cd33*	1	0.974	1.026	1
	*Cd36*	1	0.644	1.553	1
	*Msr1*	1	0.525	1.906	1
**AgNP 5 μg mL** ^**−1**^	*Ager*	0.913	0.809	1.031	0.363
	*Cd33*	0.839	0.756	0.932	0.093
	*Cd36*	1.041	0.653	1.658	0.919
	*Msr1*	0.993	0.514	1.92	0.99
**AgNP 50 μg mL** ^**−1**^	*Ager*	0.959	0.791	1.163	0.757
	*Cd33*	0.903	0.826	0.987	0.177
	*Cd36*	1.327	0.804	2.189	0.504
	*Msr1*	0.988	0.504	1.937	0.983
**CdTeQD 0.1 μg mL^−1^**	*Ager*	0.897	0.577	1.393	0.713
	*Cd33*	**0.800**	0.732	0.875	**0.040**
	*Cd36*	1.393	0.879	2.207	0.419
	*Msr1*	0.997	0.478	2.079	0.996
**CdTeQD 10 μg mL** ^**−1**^	*Ager*	0.849	0.686	1.05	0.315
	*Cd33*	**1.168**	1.107	1.232	**0.022**
	*Cd36*	**3.781**	2.397	5.963	**0.022**
	*Msr1*	1.664	0.891	3.105	0.382

Statistically significant effects (P < 0.05) are given in boldface.

**Table 2 materials-13-03227-t002:** Mean signal intensity of Aβ, AgNP and CdTeQD uptake by BV-2 cells pretreated (or not) with polyinosinic acid (PA).

Treatment		Parameter Measured (Arbitrary Units)	
		SSC (Side Scatter Value)	Fluorescence
Aβ	−PA	Not applicable	34,016 ± 5567
	+PA	Not applicable	207 ± 35.4 ^a^
AgNP	−PA	169 ± 17.6	Not applicable
	+PA	46 ± 2.9 ^a^	Not applicable
CdTeQD	−PA	Not applicable	864 ± 136
	+PA	Not applicable	1337 ± 599

Means ± SD, *n* = 3. ‘^a^’: denotes statistically significant difference between cells treated or not treated with PA, *p* < 0.05 (paired *t*-test), SSC (side scatter value).

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
