# Peer review of "The Impact of Ag Nanoparticles and CdTe Quantum Dots on Expression and Function of Receptors Involved in Amyloid-β Uptake by BV-2 Microglial Cells"

_materials, 2020, doi:10.3390/ma13143227_

Round 1

Reviewer 1 Report

Within the manuscript “The impact of Ag nanoparticles and CdTe quantum dots on expression and function of receptors involved in amyloid-β uptake by BV-2 microglial cells.” by Sikorska K. et al., the authors sought to shed light on the role of AgNPs and CdTeQDs on the expression of microglial cell surface receptors (scavenger receptors) involved in amyloid-β (Aβ) clearance.

The authors showed that the treatment of AgNPs decrease the overall content of cell membrane receptors (especially Msr1 receptor), while CdTeQDs lead to an overexpression of Cd36 receptors. Overall, the results demonstrated that while the presence of CdTeQDs may enhance inflammation, AgNPs compete with the uptake of Aβ by microglial cell, thus impairing the removal of Aβ by such cells.

Unfortunately, the results described within the manuscript are not outstanding and meaningful to the overall goal of finding new treatment of Alzheimer’s disease (plaque formation and accumulation). For this reason, I cannot recommend the manuscript for publication in Materials.

Besides, my other concerns are:

  1. Paragraph 2.3 (lines 85-86) – the authors wrote mL instead of µL
  2. Paragraph 2.5 (line 105) – FCS or FBS?
  3. Paragraph 3.1 (line 166) - the authors wrote lizosomes instead of lysosomes.

Besides, Figures 4-5 show co-localization of Aβ-HiLyte Fluor488 with mitochondria and lysosomes, respectively. I would expect to see digital images at higher magnification to better appreciate the co-localization.

My last concern is about the quantification of membrane (scavenger) receptors in microglial cells. In particular, to me, it is not clear how the authors calculate the amount of such proteins as this number is given as a percentage. Did the authors take into account the potential cytotoxic effect of AgNPs? It is not clear if the decrease of receptor amount could be due to the overall decrease of cell number (because, as the authors stated, AgNPs are likely toxic for microglial cells).

Reviewer 2 Report

Overall the manuscript by Sikorska et al is interesting, timely and well-written. There are some figure additions that would greatly enhance the paper as well as some subtle changes in the language used. 

Concerns

Fig 3 - AgNPs need an early stage and "before" panel to compare uptake

Table 1 and Fig 7- please show staining for these receptors on the cell surface to demonstrate without doubt that there are changes - flow cytometry is compelling but actual visual representation would be better

Fig 6 - Images are needed to show what precisely is being quantified in these graphs and examples of the different conditions are required. 

use of the word "proved" - this is not an appropriate phrase "demonstrated" or other synonyms of lesser strength would be more appropriate.  

Reviewer 3 Report

The manuscript entitled “The impact of Ag nanoparticles and CdTe quantum dots on expression and function of receptors involved in amyloid-β uptake by BV-2 microglial cells.” by Katarzyna Sikorska and group put forward the importance of silver nanoparticles (AgNPs) and cadmium telluride quantum dots (CdTeQDs) on the expression of Msr1, Ager, Cd36 receptors which are related to Aβ uptake by microglial cells. The work is novel, very well-drafted and explores the new possibility for the pathogenesis and treatment of Aβ mediated Alzheimer’s like condition. However, the manuscript can be accepted but with some minor corrections which will definitely help improve the quality of work.

Minor Comments:

1. The authors have studied Msr1, Cd36, Ager and Cd33 gene expression of BV-2 cells using RT-PCR but do not show any information on primers used for the RT-PCR reaction. Primer information should be added in revision as a table or in running text in the relevant section.

2. Line 283: “As already mentioned, Cd36 receptor in involved in Aβ uptake…..” should be “As already mentioned, Cd36 receptor is involved in Aβ uptake…”

Reviewer 4 Report

The manuscript entitled "The impact of Ag nanoparticles and CdTe quantum dots on expression and function of receptors involved in amyloid-β uptake by BV-2 microglial cells." by Sikorska et al. investigates the effect of silver nanoparticles (AgNPs) and cadmium telluride quantum dots (CdTeQDs) on the expression of Msr1, Ager, Cd36 receptors and Cd33 protein. The authors report that AgNPs uptake competes with Aβ uptake by microglial cells thereby impairing the removal of the aggregates. The results obtained are reasonable and the protocols used are known in the field, the manuscript should be accepted for publication after the comments below have been addressed.

  1. Although the manuscript focuses on only Aβ formation, the authors should still include information that formation of neurofibrillary tangles is also a hallmark of Alzheimer’s disease.
  2. The authors should please add the city in addition to the country for the sources of reagents and equipment used in the study.
  3. The statement “Banca Biologica e Cell Factory Centro di Risorse Biologiche IRCCS Azienda Ospedaliera Universitaria GENOVA” is not written in English, please make the correction to English.
  4. Please add source (city and country) for Nikon Eclipse Ti, Aβ-HiLyte Fluor 488, microscope Nikon A1, and ReliaPrepTMRNA Cell Miniprep System (Promega).
  5. The authors should please write FCS in full the first time it is used.
  6. In line 254, the statement “Since all receptor proteins were affected the mechanism must be general for cell membrane proteins” the authors should insert a comma between “affected” and “the”.
  7. In line 255, the statement “Although we did not tested it” should be “Although we did not test it”.
  8. In line 290, “Alzheimer Diseases” should be “Alzheimer’s Disease”.
  9. The year and volume for reference 1 are not in bold and italics, respectively.
  10. The year for reference 39, 40, and 41 are not in bold.
  11. Please remove the doi in reference 41 and 47 for consistency with other references.

added:

The few comments I had are that they should mention that formation of neurofibrillary tangles is also a hallmark of Alzheimer’s disease in the introduction. Also, they should discuss the results obtained for AgNPs and CdTeQDs in regard to developing new treatments for Alzheimer's disease.

Round 2

Reviewer 1 Report

The authors answered to all my points. In this revised form, the paper deserves to be published in Materials.